# Actin–Microtubule Crosstalk Imparts Stiffness to the Contractile Ring in Fission Yeast

**DOI:** 10.3390/cells12060917

**Published:** 2023-03-16

**Authors:** Kimberly Bellingham-Johnstun, Zoe L. Tyree, Jessica Martinez-Baird, Annelise Thorn, Caroline Laplante

**Affiliations:** 1Molecular Biomedical Sciences Department, College of Veterinary Medicine, North Carolina State University, Raleigh, NC 27607, USA; 2Quantitative and Computational Developmental Biology Cluster, North Carolina State University, Raleigh, NC 27607, USA

**Keywords:** cytokinesis, mitosis, microtubules, mechanical properties, contractile ring, quantitative microscopy

## Abstract

Actin–microtubule interactions are critical for cell division, yet how these networks of polymers mutually influence their mechanical properties and functions in live cells remains unknown. In fission yeast, the post-anaphase array (PAA) of microtubules assembles in the plane of the contractile ring, and its assembly relies on the Myp2p-dependent recruitment of Mto1p, a component of equatorial microtubule organizing centers (eMTOCs). The general organization of this array of microtubules and the impact on their physical attachment to the contractile ring remain unclear. We found that Myp2p facilitates the recruitment of Mto1p to the inner face of the contractile ring, where the eMTOCs polymerize microtubules without their direct interaction. The PAA microtubules form a dynamic polygon of Ase1p crosslinked microtubules inside the contractile ring. The specific loss of PAA microtubules affects the mechanical properties of the contractile ring of actin by lowering its stiffness. This change in the mechanical properties of the ring has no measurable impact on cytokinesis or on the anchoring of the ring. Our work proposes that the PAA microtubules exploit the contractile ring for their assembly and function during cell division, while the contractile ring may receive no benefit from these interactions.

## 1. Introduction

Actin and microtubule cytoskeletons are often considered and studied as distinct systems with individual functions. Yet, in many cellular processes, namely cell division, these two networks of polymers physically interact, and their functions combine [1,2,3,4]. Proteins or protein complexes that bind to both actin filaments and microtubules can physically connect the two cytoskeletons and modulate these interactions in space and time [4]. Physical connections between actin filaments and microtubules can align the two sets of polymers within the cell, thus providing an overall functional polarity of the combined network [5,6]. The crosslinking of stiff microtubules to flexible actin filaments may also alter their respective mechanical properties and thus modify the functions of each network. How these emergent properties of the combined actin–microtubule network impact cellular functions remains unclear.

During cell division, microtubules build the mitotic spindle while the actin network assembles a contractile ring at the division plane. Actin–microtubule interactions during cell division are well-established in higher eukaryotes. Interactions between the cortical actin network that lines the inside of the plasma membrane and the spindle microtubules position the spindle within the cell, while the spindle determines the location of the assembly of the contractile ring [1,2,7,8]. In contrast to this rich knowledge, we have limited understanding of the role and impact of actin–microtubule interactions during cell division in fission yeast, a popular model organism for studying cell division. In fission yeast, the post-anaphase array (PAA) of microtubules polymerize from specialized equatorial microtubule organizing centers (eMTOCs) connected to the actin network of the contractile ring during cytokinesis [9,10,11,12,13]. Mto1p and its dimerization partner Mto2p, gamma-tubulin complex linker proteins, and important components of MTOCs, promote the polymerization of different microtubule networks throughout the cell cycle. The recruitment of Mto1p to the contractile ring depends on the presence of myosin-II Myp2p and the activation of both the Septation Initiation Network and the Anaphase Promoting Complex [10,13]. How Myp2p enables the recruitment of Mto1p to the contractile ring remains unclear. The two proteins co-immunoprecipitated from whole-cell extract, suggesting that they are somehow connected within the same protein environment [10]. Two hybrid yeast experiments suggested that parts of the two proteins can interact directly, but whether this interaction occurs in live fission yeast cells is unknown. Importantly, the physical interactions between the stiff PAA microtubules and the more flexible actin filaments may impact the mechanical properties of the contractile ring and affect cytokinesis.

The PAA microtubules have different proposed functions, including anchoring the contractile ring to the plasma membrane/cell wall and positioning the separated daughter nuclei after anaphase [9,12,14]. In previous work, we found that defects to the mechanism that anchors the contractile ring to the plasma membrane/cell wall caused by depleting Cdc15p or deleting Imp2p, both F-BAR domain containing proteins, impact the mechanical properties of the contractile ring by decreasing the overall stiffness of the ring [15,16]. Additionally, those cells exhibited contractile rings positioned off the cell center, rings that slide along the cell after their assembly, and rings that constrict more slowly. Therefore, if PAA microtubules also anchor the contractile ring to the plasma membrane/cell wall, then they may influence cytokinesis. Alternatively, the PAA microtubules may utilize the contractile ring for their assembly as a positional scaffold without any functional gain for the contractile ring. 

Here, we combined quantitative imaging and laser ablation to determine the impact of the PAA microtubules on the contractile ring and cytokinesis. We found that the PAA microtubules form a dynamic polygon of crosslinked microtubules inside the contractile ring with bundles escaping this polygon and extending toward the cell ends. Each side of the polygon is composed of bundles of ~4 microtubules crosslinked partly by the antiparallel crosslinker Ase1p, the PRC1 homolog in fission yeast. We leveraged mutants in the *mto1* gene to interrogate the role of PAA microtubules on the mechanical properties of the contractile ring and on the anchoring of the ring to the plasma membrane/cell wall. We found that the PAA microtubules impact the mechanical properties of the contractile ring by providing stiffness to the ring during constriction. Although they modify the stiffness of the contractile ring, the ring assembled and constricted normally in the absence of PAA microtubules. We found that, contrary to previously proposed models, PAA microtubules do not anchor the ring at the division plane. Our results therefore suggest that PAA microtubules exploit the contractile ring to scaffold their assembly.

## 2. Results 

### 2.1. Myp2p Initiates PAA Microtubule Assembly without Directly Interacting with Mto1p

PAA microtubules form a complex network with robust bundles of microtubules organized in the plane of the contractile ring and other thinner bundles extending away from the ring toward the cell ends. We named equatorial PAA (ePAA) microtubules the microtubules in the plane of the ring and longitudinal PAA (lPAA) microtubules those that extended toward the cell ends. To understand how this network forms, we measured the timing of events that led to their assembly, starting with the appearance of mCherry-Myp2p (or mEGFP-Myp2p), the myosin-II essential for the recruitment of Mto1p/eMTOCs to the contractile ring (Figure 1A,B) [10]. We used the appearance of Myp2p in the contractile ring as a temporal fiduciary marker to mark time zero. The separation of the spindle pole body (SPB) occurred 18 ± 3 min before Myp2p appeared at the contractile ring (mean ± standard deviation, n = 20 cells) [10,17,18]. Shortly after the recruitment of Myp2p, at 2 ± 1 min (n = 16 cells), we detected Mto1p at the contractile ring, consistent with its dependence on Myp2p for its recruitment. The ePAA microtubules then appeared in the plane of the contractile ring soon after the arrival of Mto1p, at 4 ± 2 min (n = 20 cells). At 10 ± 3 min, ~6 min after the appearance of the ePAA microtubules, the lPAA microtubules extended from the contractile ring toward both ends of the cell (n = 20 cells). Our timing measurements support the importance of Myp2p as the initial factor that triggers the organization of the eMTOCs through the recruitment of Mto1p to the ring and the following assembly of the PAA microtubule network [10]. 

Although the recruitment of Mto1p depends on Myp2p, how Myp2p recruits Mto1p to the contractile ring is unclear. Myp2p and Mto1p may interact directly or indirectly within a protein complex, or alternatively, Myp2p may facilitate the recruitment of Mto1p without their specific co-assembly. When we imaged cells co-expressing mCherry-CHD (Calponin Homology Domain actin marker) and Mto1p-GFP or mCherry-Myp2p and Mto1p-GFP, Mto1p appeared punctate (Figure 1C,D). To determine the extent of the colocalization of Mto1p and Myp2p around the contractile ring, we acquired timelapse micrographs of cells co-expressing mCherry-Myp2p and Mto1p-GFP held upright in yeast motels, holders designed to fit fission yeast cells standing on their end [18]. These microdevices allow us to image the entire ring in face view providing a twofold improvement in resolution over reconstructed rings from stacks of focal images (Figure 1D,E and Appendix A). When imaged in yeast motels, we observed the horseshoe distribution of Myp2p during constriction that often closed into a more continuous ring as the ring constricted. The mCherry-Myp2p signal typically remained uneven around the ring and concentrated on one side of the ring throughout constriction. The Mto1p-GFP signal was uneven around the ring, comparable to mCherry-Myp2p, and it colocalized well with mCherry-Myp2p, albeit not perfectly. We frequently observed sections of the ring with mCherry-Myp2p that had no detectable Mto1p-GFP. Importantly, the Mto1-GFP signal accumulated inside the mCherry-Myp2p ring (Figure 1D,E). The mean distance between the peak fluorescence intensity of mCherry-Myp2p and Mto1p-GFP was 130 ± 50 nm (Figure 1E) (n = 6 rings). Our observations suggest that Mto1p and Myp2p co-accumulate around the ring, with Mto1p enriched in an inner layer of the contractile ring compared to Myp2p. 

Our previous observations on the imperfect colocalization of Myp2p and Mto1p suggested that Myp2p and Mto1p do not bind directly into a protein complex. To test the potential interaction between Myp2p and Mto1p, we doubled the amount of Myp2p in the contractile ring by integrating a second copy of mEGFP-Myp2p controlled by the *Pmyp2* endogenous promoter into the *leu1* locus in cells that have their endogenous Myp2p labeled with mEGFP (Figure 1F) (see Methods for details). Those cells, named Myp2p^2x^, showed a doubling of the total amount of the mEGFP-Myp2p signal in the ring (Figure 1F,G). We hypothesized that if Myp2p recruits Mto1p directly or indirectly into a protein complex, then doubling Myp2p would result in the increase of Mto1p in the contractile ring. We measured that half of the total amount of Mto1p localizes to the ring in wild-type cells, while the other half remains cytoplasmic (n = 30 cells). There is therefore sufficient Mto1p available for recruitment to the ring in Myp2p^2x^ cells should Myp2p interact with Mto1p. We compared the intensity of Mto1p-mCherry in the rings of wild-type and Myp2p^2x^ cells and found no significant change in the amount of Mto1p-mCherry recruited to the ring between those two strains (Figure 1F,H, *p* = 0.88). Together, our results propose that Mto1p accumulates at the inner face of the contractile ring in a Myp2p-dependent manner, and while Myp2p facilitates the recruitment of Mto1p, the two proteins may not interact into a protein complex.

### 2.2. PAA Microtubules Form a Polygon of Ase1p Crosslinked Microtubules inside the Contractile Ring

Following the recruitment of Mto1p to the ring, the PAA microtubules polymerized and appeared to form a ring associated with the cytokinetic contractile ring (Figure 2A). We acquired a timelapse of face views of the PAA microtubule network in cells expressing GFP-Atb2p and mCherry-Myp2p held in yeast motels (Figure 2B) [18]. When imaged face down, we observed that the ePAA microtubules organized into an irregular polygon on the inside of the mCherry-Myp2p signal. The GFP-Atb2p labeled microtubules typically concentrated where the mCherry-Myp2p signal was enriched. The sides of the microtubule polygon ran along the inside of the actin ring and connected at vertices positioned around the circumference of the mCherry-Myp2p ring (Figure 2B–D). The ePAA microtubules were dynamic, and the entire polygon reorganized on a minute timescale. These observations, along with the distribution of Mto1p, suggested that microtubules polymerize preferentially where Myp2p, and Mto1p/eMTOCs, are enriched. Presumably, the eMTOCs that nucleate the microtubules that form the sides of the polygon localize at the vertices that connect the sides of the polygon. 

Dynamic lPAA microtubules often appeared continuous across the ePAA polygon. The distal ends of the lPAA bundle showed independent dynamic instability characterized by moments of polymerization followed by sudden catastrophe (Figure 2A,E). The growth and shrinkage rates of the distal ends of the lPAA microtubules were 51 ± 20 nm/s and 96 ± 35 nm/s, similar to the range measured previously, suggesting that the +tips of these microtubules localize at their distal ends [19]. Consistently, the microtubule +tip marker Mal3p-mEGFP, the EB1 homolog in fission yeast, localized to the distal tips of the lPAA microtubules (Appendix A). We measured that each lPAA is a bundle of ~3 microtubules (2.9 ± 0.7 microtubules, n = 20 cells, see Methods). Our observations suggest that the lPAA microtubules are composed of bundles of microtubules organized with their minus ends at the eMTOCs in the ePAA/contractile ring and their plus tips pointing toward the cell ends. 

We measured that the sides of the ePAA polygon are bundles of ~4 microtubules (4.4 ± 0.6 microtubules, n = 21 cells) at their brightest. These measurements suggest that microtubules polymerize from different eMTOCs around the ring and become crosslinked into the ePAA polygon. Ase1p, the PRC1 homolog in fission yeast, preferentially bundles antiparallel microtubules and is a potential candidate crosslinker that may organize the PAA microtubule network [20,21,22,23,24]. To test this hypothesis, we determined the localization and measured the timing of recruitment of Ase1p-mEGFP to the division plane. Sparse and dim spots of Ase1p-mEGFP first appeared at the ePAA polygon at 5 ± 2 min (n = 16 cells), ~1 min after the appearance of ePAA microtubules and before lPAA microtubules appeared (Figure 1A, Figure 2F,G and Appendix A). The Ase1p-mEGFP spots became more numerous and brighter throughout the PAA microtubule network in cells with more mature constricting rings (Figure 2F,G). We hypothesized that if Ase1p was an important crosslinking factor for the network of PAA microtubules, then the bundles of the PAA microtubule network would separate and appear dimmer in the absence of Ase1p. In *∆ase1* cells, the PAA microtubule network appeared to have more lPAA microtubules extending from the ePAA microtubules at the division plane (Figure 2H,I). The mean number of lPAA microtubules per cell observed over a 5-min period was 14 ± 4 in *∆ase1* (n = 9 cells) compared to 8 ± 2 in wild-type cells (n = 11 cells). In addition, the ePAA microtubule bundles appeared dimmer, suggesting that they contained fewer microtubules (Figure 2I). We measured a decrease in the mean fluorescence intensity of ePAA microtubules of 11% in *∆ase1* cells (n = 30 ePAA microtubules) compared to wild type cells (n = 30 ePAA microtubules). Our results suggest that Ase1p is one of the crosslinkers responsible for bundling the PAA microtubules into a network, and it controls the number of lPAA microtubules that escape from the polygon of ePAA microtubules.

### 2.3. eMTOCs Impact the Mechanical Properties of the Contractile Ring

The PAA microtubules may anchor the contractile ring at the division plane [9]. In previous work, we found that depleting or deleting proteins that anchor the contractile ring resulted in changes in the mechanical properties of the ring [15,16]. Based on these works and on the different properties of microtubules and actin filaments, the physical interactions between PAA microtubules and the contractile ring may affect the mechanical properties of the ring. To determine whether and how the PAA microtubules impact the mechanical properties of the ring, we used laser ablation to sever the contractile ring in wild-type cells and in cells that lack PAA microtubules (*∆mto1* and *mto1-427* cells) (Figure 3A and Appendix A) [10,25]. The complete loss of Mto1p in *∆mto1* cells impacts all microtubule networks in the cell, while *mto1-427* specifically affects PAA microtubules (Figure 3A) [10]. Severing the constricting contractile ring releases the tension present in the ring prior to ablation, causing the severed tips of the contractile ring to recoil away from the site of ablation [15,16,26]. We used mEGFP-Myp2p as a contractile ring marker and to track the dynamics of the severed tips [15,16]. We imaged each cell prior to laser ablation, then severed their contractile ring and imaged the ablated contractile ring every second for 5 min (Figure 3B) (see Methods for details). We imaged a single focal plane of the surface of the contractile ring after ablation to achieve high temporal resolution while preventing photobleaching (Appendix A). Only contractile rings that were 20–50% constricted were used in the analysis, as rings that have not started to constrict may have distinct mechanical properties. In all three genotypes, the severed tips recoiled, the recoiling stopped, and then the severed rings healed (Figure 3B). The majority of severed rings (89%) in both wild-type and *mto1-427* cells healed during our 5 min acquisition period. However, ~30% of the severed *∆mto1* rings failed to heal by the end of our acquisition period, suggesting that cells that completely lack Mto1p have a mild deficiency in healing gaps in their contractile rings. This healing phenotype is likely due to other defects besides the lack of PAA microtubules, as it was not observed in *mto1-427* cells. 

We tracked the position of the severed tips during the recoil phase and calculated their recoil displacement (Figure 3C). As expected for a material with viscoelastic properties, the mean recoil displacement of the severed tips ΔL measured in wild-type, *∆mto1*, and *myo1-427* cells fitted to an exponential, $\Delta L\left(t\right)=A\left(1-e^{-\frac{t}{\tau }}\right)$, [15,16]. For severed rings in wild-type cells, we measured the magnitude of the recoil after ablation, A
*=* 0.45 ± 0.006 µm (mean ± standard error on the least squares fit, SE), and a timescale of viscoelastic recoil, *τ =* 24 ± 0.8 s (mean ± SE). The recoil magnitude and timescale of the displacement profile of the severed tips increased in both *∆mto1* and *mto1-427* cells compared to wild-type cells (A = 0.52 ± 0.0006 µm for *∆mto1* and A= 1.03 ± 0.040 µm for *mto1-427*) (*τ* = 20 ± 0.7 s for *∆mto1* and *τ* = 58 ± 3.6 s for *mto1-427*) (Figure 3C). The experimental displacement curves for *∆mto1* and *mto1-427* cells did not reach a plateau under our experimental conditions due to some severed tips recoiling out of the imaging plane or healing before the recoil phase reached a plateau (Appendix A). Therefore, because our data did not reach a plateau, our reported values of *A* and *τ* for mutant strains represent lower limits on these responses. 

We calculated the impact of Mto1p on the effective stiffness k and effective viscous drag η of the contractile ring based on our framework for the contractile ring as a viscoelastic material under tension $T_{0}$ [16,26]. The effective stiffness of the ring represents the sum of the stiffness of all the molecular components of the ring. Similarly, the effective viscous drag of the ring represents the sum of the viscous drag of all the molecular components of the ring. Both A and τ describe the displacement profile of the severed tips and are influenced by the mechanical properties of the contractile ring as $A=\frac{T_{0}}{k}$ and $\tau =\frac{\eta }{k}$ (Figure 3C). Given the relationship between A and k, the increased A, and assuming that $T_{0}$ is unchanged in *∆mto1* and *mto1-427* cells, our data suggests that the k of the contractile ring decreases in both mutant strains. In addition, Mto1p likely impacts k more than η, as A and *τ* increased by a similar factor and both A and *τ* rely on k. Our measurements suggest that the lack of Mto1p in the contractile ring primarily affects the stiffness of the contractile ring. Based on their molecular organization and connection to the actin ring, the lack of Mto1p, eMTOCs, and PAA microtubules likely all contribute to the decrease in the effective stiffness of the contractile ring in *mto1-427* and *∆mto1* cells. Although *∆mto1* cells have more severe microtubule defects, they exhibited a milder impact on the mechanical properties compared to *mto1-427*, which only lack PAA microtubules. As the microtubule cytoskeleton influences many cellular functions, the cumulative defects of *∆mto1* cells are likely responsible for the differences we measured in recoil displacement between *∆mto1* and *mto1-427*. The impact of *mto1-427* on the mechanical properties of the contractile ring is likely more specific to the loss of the PAA microtubule. The impact of the PAA microtubules on the effective stiffness of the ring is consistent with the potential role of PAA microtubules in the general anchoring mechanism of the contractile ring. 

### 2.4. The Lack of PAA Microtubules Has No Impact on Cytokinesis

Depleting or deleting proteins that anchor the contractile ring to the plasma membrane/cell wall caused a decrease in the effective stiffness of the contractile ring and in the reduction of the constriction rate of the ring [15,16]. These results suggest that tension force applied to a ring with lower stiffness results in a slower rate of constriction, perhaps owing to the poor transmission of the tension force to the plasma membrane/cell wall. The impact of PAA microtubules on the effective stiffness of the contractile ring suggests that the loss of PAA microtubules may alter the function of the contractile ring. We measured the timing of cytokinetic events in wild-type, *∆mto1*, and *mto1-427* cells expressing mEGFP-Myo2p as a marker of the mitotic nodes and contractile ring and Sad1p-RFP to label the SPB (Figure 4A,B). We used SPB separation as time zero. In wild-type cells, a band of mitotic nodes, complexes of cytokinetic proteins, appeared at the division plane at −5 ± 2 min (n = 20 cells) [17,18]. As measured previously, the nodes started to coalesce into a continuous ring coincident with SPB separation, and the ring was fully formed by 20 ± 3 min (n = 20 cells). After a brief maturation phase, the ring began to constrict at 30 ± 4 min (n = 20 cells) and was fully disassembled at 66 ± 4 min (n = 20 cells) once constriction ended. In both *∆mto1* and *mto1-427* cells, all cytokinesis events occurred with approximately the same timing as in wild-type cells with one exception. In *∆mto1* cells, the onset of ring constriction was slightly delayed, as rings began to constrict at 38 ± 6 min (n = 20 cells) and rings disassembled later than in wild-type cells at 71 ± 5 min (n = 20 cells), consistent with the delayed onset in constriction. Therefore, the lack of PAA microtubules alone had no measurable impact on the timing of cytokinesis. The mild timing delays measured in *∆mto1* cells are likely due to the general impact of the lack of Mto1p on all microtubule networks of the cell. 

We measured the constriction rate of mEGFP-Myo2p labeled contractile rings in wild-type, *∆mto1*, and *mto1-427* cells (Figure 4C). The rates of ring constriction in wild-type and *mto1-427* cells were comparable (0.28 µm/min for wild-type and 0.27 µm/min for *mto1-427*) (n = 20 cells for each). The rate of ring constriction was slightly faster in *∆mto1* cells at 0.33 µm/min (*p* < 0.005, n = 18 cells). In *∆mto1* cells, the constricted ring lingered at the end of constriction before fully disassembling (Figure 4D). Therefore, although the loss of PAA microtubules caused a decrease in ring stiffness, the change in the mechanical properties of the ring did not affect cytokinesis in measurable ways.

### 2.5. PAA Microtubules Alone Do Not Anchor the Contractile Ring

Previous work suggested that the PAA microtubules anchor the contractile ring at the division plane [9]. Pardo and Nurse showed that cells carrying a mutation in the cell wall synthesis enzyme Bgs1p/Cps1p, *cps1-191* cells, treated with carbendazim/MBC, a drug that depolymerizes microtubules, exhibited contractile rings positioned off-center. In our previous work, we found that mutations that weaken the anchoring mechanism of the contractile ring lowered the effective stiffness of the contractile ring [15,16]. Based on those studies, we expected cells that lack PAA microtubules to show contractile rings positioned off-center and a ring sliding phenotype, hallmark phenotypes of the defective anchoring of the ring to the plasma membrane/cell wall [27,28]. We measured the position of the contractile ring at the time of constriction onset in wild-type, *mto1-427*, and *∆mto1* cells. Both wild-type and *mto1-427* mutant cells had rings positioned at the cell center (Figure 5A) [10]. However, 28% of *∆mto1* cells had contractile rings positioned off-center (>10% offset from the calculated cell center). We acquired timelapse micrographs of wild-type, *∆mto1*, and *mto1-427* cells and determined whether their contractile ring slid away from their position of assembly or assembled at a position away from the cell center. We measured ring sliding as the difference in the position of the ring between the time of assembly and the time of constriction onset. We measured no ring sliding in wild-type, *∆mto1*, and *mto1-427* cells, and rings in those cells remained where they assembled (Figure 5B,C). Our results suggest that PAA microtubules do not impact the anchoring of the contractile ring at the division plane.

In fission yeast, the contractile ring assembles at the cell center, and the position of the nucleus instructs where the contractile ring assembles [29,30]. We measured the position of the ring at the time of assembly in *∆mto1* cells and found that they frequently assembled off-center (Figure 5D). Consistent with the off-center position of the contractile ring at the time of ring assembly, the nucleus was often positioned off-center at the time of ring assembly in *∆mto1* cells (Figure 5E). Therefore, the nucleus is positioned off-center in *∆mto1* cells, causing the assembly of their contractile ring to be off-center. 

To shed light on the nuclear positioning defect measured in *∆mto1* cells, we observed nuclear displacement in these cells. Timelapse micrographs of *∆mto1* cells often showed exaggerated motions of the nucleus in interphase cells (Appendix A). In wild-type cells, interphase microtubules center the nucleus prior to the assembly of the contractile ring [29,30,31]. Timelapse micrographs of wild-type and *mto1-427* cells during interphase showed the nucleus oscillating gently around the cell center (Appendix A) [30,31]. In contrast, the nucleus in *∆mto1* cells often explored nearly the entire length of the cell. In both wild-type and *mto1-427* cells, nuclei were mostly circular. Transient deformations caused by the pushing actions of the microtubules on the nuclear envelope at the points of attachments transiently deformed the nuclear envelope in those cells (Appendix A) [30,31]. Nuclei in *∆mto1* cells sometimes adopted an elongated lemon shape (Figure 5F and Appendix A). As the attachment of microtubules to the nuclear envelope can alter the shape of the nucleus, we observed interphase microtubules in *∆mto1* cells. Wild-type and *mto1-427* cells had three or four interphase microtubules, while *∆mto1* cells had zero to two interphase microtubules (Figure 5G) [10]. Only cells that had a single interphase microtubule exhibited the lemon-shaped nuclear phenotype (n = 15 cells). In those cells, the nuclear envelope was deformed where it contacted the single interphase microtubule, and this deformation made the nucleus appear lemon-shaped. Therefore, the nucleus in *∆mto1* cells is not stably positioned at the cell center, possibly due to their aberrant interphase microtubules causing the exaggerated nuclear motions. Together, our observations suggest that *∆mto1* cells that either lack interphase microtubules or have a single interphase microtubule have increased chances of having an off-center nucleus at the time when the position of the division plane is determined. Importantly, these results support that the PAA microtubules do not anchor the contractile ring to the division plane. 

## 3. Discussion 

The mechanics and function of actin and microtubule polymers have been studied independently for years. Our best understanding of these molecules reflects their individual isolated characteristics. In live cells, however, actin and microtubules frequently interact. These physical interactions alter their mechanical properties and influence their cellular functions [4,6]. In fission yeast, the microtubule and actin cytoskeleton physically interact during cytokinesis. We know best about cytokinesis in fission yeast owing to years of genetics, cell biology, and mathematical modeling experiments. Recently, we began to elucidate the molecular organization of the contractile ring and its mechanical properties [15,16,26,32,33,34]. Such extensive understanding of cytokinesis in fission yeast enables us to dig into the impact of the actin–microtubule crosstalk in cytokinesis in vivo. Here, we used fission yeast to investigate the impact of PAA microtubules on the actin contractile ring and cytokinesis. We leveraged a particular mutation in *mto1*, *mto1-427*, to specifically eliminate PAA microtubules and test the outcome on cytokinesis. Our timing analysis supports the sequential assembly of the PAA microtubule following Myp2p recruitment to the contractile ring, with Myp2p arriving to the ring first, rapidly followed by Mto1p/eMTOCs, PAA microtubules, and eventually Ase1p crosslinkers. This progressive assembly is consistent with the previous observation that the actin network and Myp2p are required for Mto1p and eMTOC recruitment to the contractile ring [10,13]. 

Co-immunoprecipitation from whole cell extract and yeast two-hybrid experiments suggested that Myp2p and Mto1p assemble into a protein complex and may interact directly. Our results, however, argue against the direct interactions of Mto1p and Myp2p in vivo for two reasons. First, although Myp2p and Mto1p co-accumulated unevenly around the ring, the two proteins did not perfectly co-localize. Instead, Mto1p accumulated on the inner face of the Myp2p signal around the ring, with their peaks in fluorescence intensity separated by ~130 nm. Mto1p was also distributed as puncta around the ring, suggesting its local accumulation, while the Myp2p signal appeared smooth with no indication of similar local clustering. Second, doubling the total amount of Myp2p did not increase the recruitment of Mto1p to the contractile ring. This result suggests that, although the presence of Myp2p is required for Mto1p recruitment [10], this recruitment mechanism is independent of the concentration of Myp2p in the ring. Together, our results propose that Myp2p facilitates the recruitment of Mto1p without forming a specific complex with Mto1p. One possible mechanism for Myp2p to facilitate Mto1p recruitment could be that Myp2p modifies the actin network in a way that enables the local recruitment of Mto1p to the inner face of the contractile ring. Alternatively, if Mto1p forms a protein complex with Myp2p, this complex would only assemble locally, on the inside portion of the Myp2p ring, which may spatially restrict the localization of Mto1p and limit the total amount of Mto1p recruitment. Future work is required to understand how Mto1p joins the ring, locally sorts to the inner face of the ring, and connects to the actin network and how Myp2p facilitates these events.

By imaging the contractile ring and PAA microtubules face down, we determined that the PAA microtubule network forms a dynamic polygon on the inside of the actin contractile ring. This is consistent with the stiffness of microtubules, characterized by their high persistence length, preventing bundles of microtubules from bending into a ring of ≤3.5 µm s in diameter [35,36]. The formation of this polygon implies some local molecular organization of the Mto1p/eMTOCs and crosslinkers. Presumably, microtubules polymerize from Mto1p/eMTOCs and are rapidly crosslinked by Ase1p into a polygon of ePAA microtubules. Microtubule bundles thus form the sides of the polygon and connect at vertices where the Mto1p/eMTOCs that polymerized those microtubules are likely enriched. Microtubules that escape the ePAA bundles elongate toward the cell ends and become lPAA microtubules. The presence of Ase1p puncta around ePAA microtubules suggests that the polygon is partly made of antiparallel bundles [20,21,22]. Deleting *ase1* resulted in an increased number of lPAA microtubules, suggesting that Ase1p crosslinking may control the ratio of lPAA to ePAA microtubules within the array. Although Ase1p participates in bundling the PAA microtubules, it may not be the sole crosslinker responsible for bundling the ePAA microtubules, as deleting *ase1* did not eliminate PAA microtubule bundles. Deleting *ase1* instead decreased the fluorescence intensity of the microtubule bundles by ~10%, suggesting that bundles of microtubules still form in the absence of Ase1p but contained fewer microtubules. Other possible crosslinkers include kinesin motors, such as Klp2p and Plk1p, and the TOG/XMAP215 homolog Dis1p, which have been implicated in the bundling of other microtubule networks in fission yeast [37,38]. 

One of the proposed functions of PAA microtubules is to position the nuclei after anaphase [12,14]. The organization and polarity of the PAA networks may provide insights into this mechanism. Connecting the minus ends of the lPAA microtubules to the contractile ring ensures that the plus ends extend away, providing a specific polarity to the network. This organization may serve to point the dynamic ends of the lPAA away from the ring toward the nuclei, allowing the +tips of the lPAA microtubules to exert pushing forces against the daughter nuclei while their –ends are connected to the ePAA polygon/contractile ring. In addition, this orientation may be necessary for the transport of material away from the ring and toward the cell ends. 

We expected the attachment of stiff bundles of microtubules to the actin contractile ring to alter the mechanical properties of the contractile ring. Severing the contractile ring revealed that the PAA microtubules indeed impact the mechanical properties of the ring, specifically by contributing to the effective stiffness of the ring. This outcome also suggested that removing the PAA microtubules would alter the function of the contractile ring and cytokinesis. However, the timing of cytokinetic events and the constriction rate of the contractile ring in cells that lacked PAA microtubules were indistinguishable from wild-type cells, suggesting that, although the PAA microtubules impart stiffness of the ring, they do not generally affect cytokinesis. These results were even more surprising given that the PAA microtubules were expected to anchor the contractile ring to the plasma membrane/cell wall [9]. Indeed, we previously found that depleting Cdc15p or deleting Imp2p, anchoring protein candidates, decreased the effective stiffness of the contractile ring and caused a decrease in the constriction rate, possibly linking ring stiffness with its constriction rate [15,16]. Pardo and Nurse concluded that the PAA microtubules anchor the contractile ring by studying cells carrying a mutation in the cell wall synthesis enzyme Bgs1p/Cps1p, *cps1-191* cells, treated with carbendazim/MBC, a microtubule depolymerizing drug [9]. These cells exhibited contractile rings positioned off-center and rings that slide away from their position of assembly. It is important to note that cells carrying the *cps1-191* mutation exhibit a ring sliding phenotype and that, in Cdc15p depleted cells, the ring slides until Bgs1p levels reach a critical threshold for anchoring the ring in place, indicating that Bgs1p/Cps1p itself acts as a contractile ring anchor [27,28,39]. Our results showed that the specific loss of PAA microtubules had no measurable effect on ring anchoring. Perhaps in a sensitized background, such as *cps1-191*, removing the microtubules may enhance an already existing anchoring defect phenotype. Pardo and Nurse performed their experiments in an anchoring defective background, and the ring sliding phenotype they observed may have been exacerbated by removing microtubules with MBC treatment. 

Both the mechanism that anchors the contractile ring to the plasma membrane/cell wall and the PAA microtubules impact the stiffness of the contractile ring. Yet, only defects in anchoring alter the function of the ring as measured by its constriction rate, suggesting that the molecular source of the stiffness may determine the impact on cytokinesis. The absence of a cytokinesis phenotype in cells lacking PAA microtubules is perplexing. One possible explanation is that cytokinesis is affected by the loss of PAA microtubules but not in a way that can be captured by measuring the rate of ring constriction. Afterall, the rate of ring constriction is likely determined by a combination of factors, including the action of the constricting contractile ring and the deposition of the cell wall [40]. As the cell wall is a critical component of the anchoring mechanism, a poorly anchored ring may also have a defective cell wall, resulting in a slower rate of ring constriction. Alternatively, the loss of PAA microtubules may result in changes to the contractile ring that are not captured by the constriction rate. For example, PAA microtubules may influence the organization of the actin filament network, as microtubules and actin filaments can mutually influence their organization and polarity in vitro [5,6]. Redundant mechanisms may then compensate for the lack of PAA microtubules to ensure a robust cytokinesis. Although the relationships between the stiffness of the contractile ring and its functions remain unclear, our work underlines that actin–microtubule crosstalk during cytokinesis in fission yeast, and possibly in other organisms, influences the mechanical properties of the contractile ring. This crosstalk is critical for the assembly and presumably the function of the PAA microtubule network, which use the ring as a positional scaffold. Whether the contractile ring also benefits from this crosstalk remains to be determined. 

## 4. Methods and Materials

### 4.1. Strains, Growing Conditions, and Genetic and Cellular Methods

Appendix A lists the *S. pombe* strains used in this study. Strains were created using PCR-based gene targeting to integrate the constructs into the endogenous locus, except for *Pnmt41-mCherry-CHD* and mEGFP-Myp2p^2x^, which were integrated into the *leu1* locus [41]. The *pFA6a-mCherry-kanMX6* vector was used as a template to generate C-terminal tagged constructs. The primers had 80 bp of homologous sequence with the endogenous genomic integration site (designed with the help of www.bahlerlab.info/resources/). Except for the *Pnmt81-GFP-atb2* and *Pnmt41-mCherry-CHD* constructs, all tagged genes were controlled by their endogenous promoter. Successful integrations were confirmed by a combination of DNA sequencing, PCR, and fluorescent microscopy.

Doubling Myp2p in the mEGFP-Myp2p^2x^ strain was accomplished by integrating an additional copy of *megfp-myp2* into the *leu1* locus, using a modified version of the pJK148 vector [42] containing 800bp of the endogenous *myp2* promoter, the full coding sequence of *megfp*, two repeats of GGAGGT to create a 4xGly linker, the full coding sequence of *myp2*, and 420bp of the endogenous *myp2* terminator. Strains were verified by PCR and by sequencing the integration site.

For imaging, cells were grown to the exponential phase for 36–48 h before imaging with regular dilutions. Strains containing the *Pnmt81-GFP-atb2* and *Pnmt41-mCherry-CHD* constructs were shifted to EMM5S 15–18 h prior to imaging to allow expression of the constructs. 

### 4.2. Spinning-Disk Confocal Microscopy 

Cells were grown to the exponential phase at 25 °C in YE5S-rich liquid medium in 50-mL baffled flasks in a shaking temperature-controlled incubator in the dark. Cells were concentrated 10- to 20-fold by centrifugation at 2400× *g* for 30 s and then resuspended in EMM5S. A total of 5 μL of cells was mounted on a thin gelatin pad made of 10 μL 25% gelatin (Sigma Aldrich; G-2500) in EMM5S, sealed under a #1.5 coverslip with VALAP (1:1:1Vaseline:Lanolin:Parafin) and observed at 22 °C. To image cells vertically in yeast holders (yeast motels), 2–5 μL of a diluted resuspension of yeast cells as described above was pipetted onto the surface of a polydimethylsiloxane mold containing pillars 6 µm in diameter and 14 µm high [18]. The mold was then inverted onto a 40 mm in diameter circular #1 coverslip, and the cells were imaged immediately. 

Fluorescence micrographs of live cells were acquired with a Nikon Eclipse Ti microscope equipped with a 100×/numerical aperture (NA) 1.49 HP Apo TIRF objective (Nikon), a CSU-X1 (Yokogawa) confocal spinning-disk system, 488/561 nm solid state lasers, and an electron-multiplying cooled charge-coupled device camera (EMCCD IXon 897, Andor Technology). The Nikon Element software was used for acquisition. 

ImageJ was used for all image visualization and analyses [43]. Images in the figures are either maximum intensity projections of z-sections spaced at 0.36 μm or single z-planes acquired at a high temporal resolution. For cells imaged lying flat on a gelatin pad, 19 z-planes (6.48 µm deep) were acquired. For cells imaged in yeast motels, 13 z-planes (4.32 µm deep) were acquired. Images were systematically contrasted to provide the best visualization, and images within the same figure panel were contrasted using the same settings. 

### 4.3. Image Analyses 

The timing of events associated with PAA microtubules (Figure 1A) was measured by noting the camera frame when the SPB separated, the appearance of mEGFP-(or mCherry-)Myp2p, Mto1p-GFP, GFP-Atb2 (ePAA and lPAA microtubules), and Ase1p-mEGFP, and the disappearance of the ePAA microtubules and the Myp2p-labeled ring. The completion of Myp2p or ePAA microtubule disassembly was defined as the first timepoint when the signal from the structure was no longer visible at the end of constriction.

To measure the fluorescence intensity in the contractile rings of wild-type and mEGFP-Myp2p^2x^ cells (Figure 1F,G), we created sum projection images of fields of cells from stacks of 21 optical images separated by 0.36 μm. The images were corrected for camera noise and uneven illumination, and then the fluorescence intensity of contractile rings was measured. A standard least squares model of the form Y = Ring Diameter + Genotype + Ring Diameter*Genotype, where Y is the fluorescence intensity, was used to determine whether the distribution of fluorescence intensity in the contractile ring differed between wild-type and mEGFP-Myp2p^2x^ cells. Tests were performed with JMP Pro 16 (SAS Institute, Cary, NC, USA). 

Contractile ring timing (Figure 4B) was measured by noting the camera frame where the SPB separated, the broad band nodes appeared, the contractile ring was fully assembled, constriction onset occurred, and the contractile ring disassembled. The completion of contractile ring assembly was defined as the first timepoint when all contractile ring material was fully incorporated into a single contractile ring structure. Constriction onset was defined as the first timepoint at which the contractile ring started to consistently constrict. The completion of contractile ring disassembly was defined as the first timepoint when the signal from the constricting contractile ring was no longer visible at the end of constriction. 

The offset of the contractile ring at the time of ring constriction (contractile ring offset) (Figure 5A) was measured using maximum-intensity projections of stacks of micrographs of cells expressing mEGFP-Myo2p. The center of the cell was calculated as half the cell length (*L*). The position of the contractile ring (*X*) was measured as the distance between the cell end and the contractile ring. The contractile ring offset was calculated using the following equation:$$Ring\;offset=\left(\frac{\left|X-\left(\frac{L}{2}\right)\right|}{\frac{L}{2}}\right)\times 100$$

To calculate the change in the ring offset (Figure 5B), we measured the position of the contractile ring at the time point when it was fully assembled (*Xa*) and the position of the contractile ring at the onset of constriction (*Xc*). We calculated the difference between the ring offset at both time points using the following equation:$$Change\;in\;ring\;offset=\left(\frac{\left(\left|Xc-\left(\frac{L}{2}\right)\right|\right)-\left(Xa-\left(\frac{L}{2}\right)\right)}{\frac{L}{2}}\right)\times 100$$

Centroid measurements (Figure 5D,E) were calculated using FIJI to obtain the *x* and *y* coordinates of the center of the cell, contractile ring, and nucleus at the time of contractile ring assembly. Rectangles were drawn around the cell and the contractile ring separately, and their center coordinates were calculated using the “centroid” option in the “set measurements” option in FIJI. Similarly, a circle was drawn around the nucleus, and its center coordinates were calculated using the “centroid” option in the “set measurements” option in FIJI. The distance between centroids was calculated using the following formula, where ΔX is the difference between the *x* coordinates and ΔY is the difference between the *y* coordinate values:(1)$$Distance\;between\;centroids=\sqrt{{\Delta X}^{2}+{\Delta Y}^{2}}$$

Student’s *t*-tests were used to determine whether the distances between object coordinates were significantly different. 

To determine the number of microtubules per bundle present in different microtubule networks, we measured the fluorescence intensity in the fully elongated spindle, ePAA microtubules at the time of spindle disassembly, and lPAA microtubules from sum projection timelapse images of fields of cells from stacks of 19 optical images separated by 0.36 μm. The images were corrected for photobleaching using the “exponential method” in the Bleach Correction ImageJ plugin [44]. The numbers of microtubules per bundle in ePAA and lPAA microtubules were calculated using the relative fluorescence intensity when compared to that of the fully elongated spindle microtubules. Fully elongated spindle microtubules were assumed to contain four microtubules based on previous measurements [45].

To measure the reduction in the number of microtubules present in the ePAA of *Δase1* cells, we measured the fluorescence intensity of the ePAA microtubules in wild-type and *Δase1* cells from the sum projections of fields of cells from stacks of 21 optical images separated by 0.36 μm. The percent reduction was calculated by comparing the relative fluorescence intensity between wild-type and *Δase1* cells. A Student’s *t*-test was used to determine whether the difference in fluorescence intensity was significantly different.

To measure the fluorescence intensity of the cytoplasmic fraction of Mto1p-mCherry in wild-type cells during cytokinesis, we created sum projections from images of fields of cells from stacks of 21 optical images separated by 0.36 μm. The images were corrected for camera noise and uneven illumination, and then the fluorescence intensity of the contractile ring and the global fluorescence intensity was measured for each cell. The fluorescence intensity of the cytoplasm was calculated as the difference in fluorescence intensity between the contractile ring and the global fluorescence intensity.

### 4.4. Ablation Analysis

The laser ablation of contractile rings and the analysis of the recoil of the severed tips were performed as described previously [15,16]. We included the detailed methods here as well. We severed constricting contractile rings in wild-type, *mto1-427*, and *∆mto1* cells expressing mEGFP-Myp2p as a ring marker. We focused on constricting contractile rings (∼80–50% of their initial size) to avoid potentially confounding results from contractile rings that did not yet start to constrict, as they may have had distinct mechanical properties. We did not sever rings that constricted by >50% because their severed tips are only visible for a short duration in the single focal plane and cannot be tracked for longer durations (Appendix A). While we were able to visualize both severed tips in a few severed contractile rings, only one tip remained in our observation plane in most cases. We ablated 27 contractile rings in wild-type cells, 18 contractile rings in *mto1-427* cells, and 21 contractile rings in *∆mto1* cells. 

The laser ablation of contractile rings was performed as described before on a Nikon Ti-E microscope equipped with an Andor Dragonfly spinning-disk confocal fluorescence microscope equipped with a 100× NA 1.45 objective (Nikon, Tokyo, Japan) and a built-in 1.5× magnifier, a 488 nm diode laser with Borealis attachment (Andor, Belfast, UK), emission filter Chroma ET525/50m, and an EMCCD camera (iXon3, Andor Technology) and a MicroPoint (Andor) system with galvo-controlled steering to deliver two sets of 15 3 ns pulses of 551 nm light at 16 Hz (Andor) [15,16]. Fusion software (Andor) was used to control the acquisition while IQ software (Andor) was used simultaneously to control the laser ablation. At this pulse rate, the ablation process lasted 1–2 s. A Chroma ET610LP mounted in the dichroic position of a Nikon filter turret was used to deliver the ablation laser to the sample. Because this filter also reflects the mEGFP emission, the camera frames collected during the ablation process were blank. The behavior of severed contractile rings was imaged immediately following laser ablation by acquiring a single confocal plane in the 488-nm channel every second for 5 min. We acquired time-lapse images of a single optical plane of the surface of the contractile ring to maximize temporal resolution and reduce photodamage. 

The displacement of the tips of severed contractile rings was tracked manually every second after laser ablation using the “multi-point” tool in FIJI on single z-plane time-lapse micrographs [16]. The severed tips were tracked until they either finished their initial period of recoil, recoiled out of the imaging plane, or until the contractile ring was healed. The coordinates of the tracked severed tips were exported as a CSV file. The displacement of the severed tips starting from the first frame after severing, t = 0 s, was calculated from the coordinates of the tracked severed tips using custom Python codes [16]. The recoil displacement traces from each cell were aligned to t = 0 s, and the mean displacement curve was calculated for each genotype. The recoil displacement traces showed the combined recoil phase of all severed tips exclusively. The recoil displacement of each of the severed tips contributed to the traces until the recoil halted. In cases where recoil stopped and restarted again, only the first recoil was used in the displacement calculation. The mean displacement curve for each genotype was fitted to a single exponential using the least squares fit, and the SE on the least squares fit was reported. Contractile rings were considered fully ablated if there was no remaining fluorescence joining the ablated tips of the ring and the ablated ends showed evidence of recoil away from the site of ablation. We interpreted the motion of the severed tips as recoil due to tension release and not due to actin depolymerization because our previous work showed that there is a local enrichment of mEGFP-Myp2p at the severed tip after ablation. This phenomenon suggests that severing the contractile ring results in the rearrangement of its proteins to meet the new mechanical constraints [16]. Although there may be some depolymerization of the actin, we believe that the motion of the severed tips is mainly caused by recoil due to tension release. 

## Figures and Tables

**Figure 1 cells-12-00917-f001:**
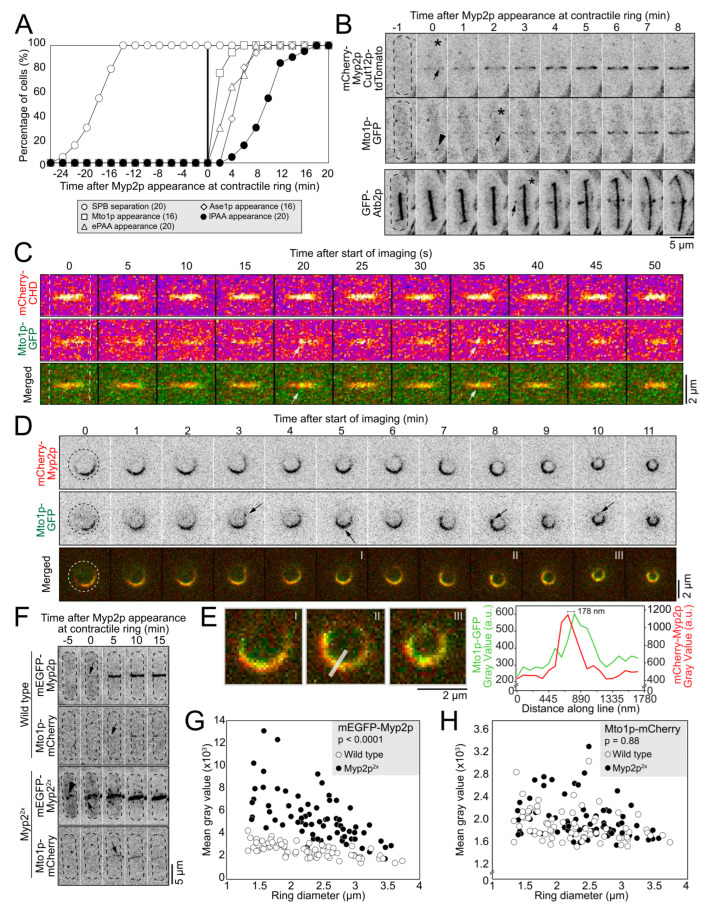
Myp2p facilitates Mto1p recruitment to contractile ring without their direct interaction. (**A**), Outcome plot showing the timing of events leading to the assembly of the post-anaphase array (PAA) of microtubules network. Time zero, appearance of Myp2p in the contractile ring. Parentheses, number of cells analyzed. (**B**), Timelapse micrographs of cells expressing mCherry-Myp2p, Cut12p-tdTomato and Mto1p-GFP, or mCherry-Myp2p and GFP-Atb2p. Top and middle rows, same cell. Only GFP-Atb2p is shown in the bottom cell. Dashed outlines, cells. Asterisks, timepoint of signal appearance. Arrows, location of signal appearance. Arrowhead, Mto1p at the SPB. (**C**), Timelapse micrographs of the surface of the contractile ring of a cell expressing mCherry-CHD (Calponin Homology Domain) and Mto1p-GFP. Dashed lines, cell edges. Arrows, puncta of Mto1p. (**D**), Timelapse micrographs of a cell expressing mCherry-Myp2p and Mto1p-GFP imaged in a yeast motel. Dashed circles, cell outlines. Arrows, puncta of Mto1p. Roman numerals, images enlarged in E. (**E**), Enlarged timepoints from D. Grey line is the line used for fluorescence intensity scan in graph. Graph, fluorescence intensity of both signals along the line. (**F**), Timelapse micrographs of cells expressing mEGFP-Myp2p and Mto1p-mCherry in wild-type (top) or Myp2p^2x^ cells (bottom). Dashed outlines, cells. Arrows, appearance of Myp2p or Mto1p in the contractile ring. Arrowhead, cytoplasmic Myp2p dots. (**G**), Plots of the distribution of the mean fluorescence intensity of mEGFP-Myp2p in wild-type and Myp2p^2x^ cells. (**H**), Plots of the distribution of the mean fluorescence intensity Mto1p-mCherry in wild-type and Myp2p^2x^ cells. Statistical significance was determined by the least squares mixed model in both G and H. Except for the C fire lookup table (LUT) and color merged images, all micrographs are shown as inverted grayscale LUT.

**Figure 2 cells-12-00917-f002:**
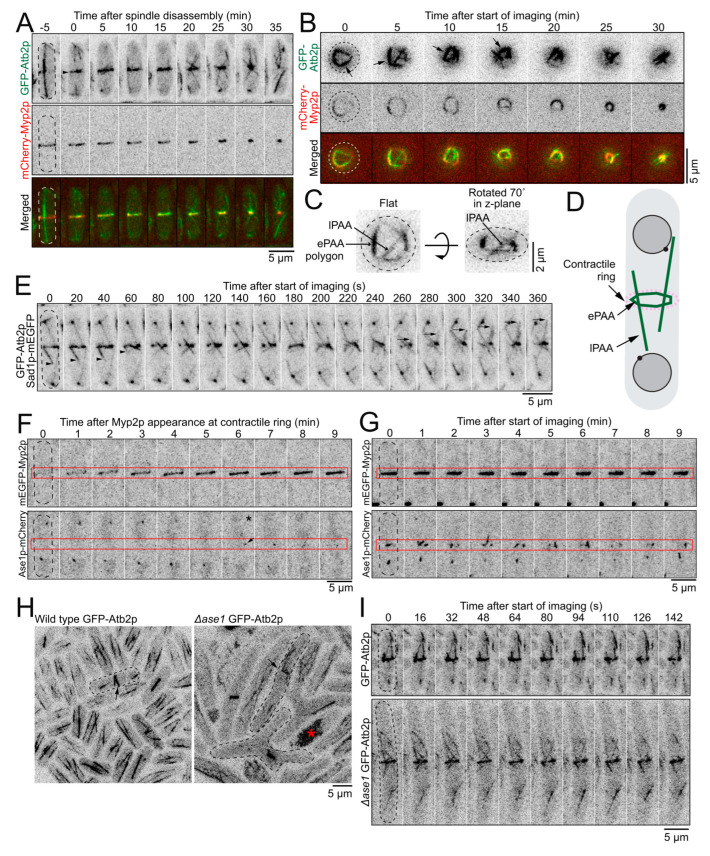
Polygon of ePAA microtubules bundled by Ase1p. (**A**), Timelapse micrographs of cells expressing GFP-Atb2p and mCherry-Myp2p. Arrowhead, ePAA microtubules in the plane of the contractile ring. Dashed outlines, cells. (**B**), Timelapse micrographs of a cell expressing GFP-Atb2p and mCherry-Myp2p imaged in a yeast motel. Arrows point to the equatorial PAA (ePAA) polygon in the first 4 timepoints. The ePAA microtubules cannot be confidently distinguished from the longitudinal PAA (lPAA) microtubules in the last 3 timepoints, owing to their overlap in projected images. Dashed circle, cell outline. (**C**), Micrograph of cell expressing GFP-Atb2p in a yeast motel (left). Same cell rotated 70 degrees around the *z*-axis (right). (**D**), Diagram depicting the organization of the polygon of PAA microtubules. (**E**), Timelapse micrographs of a cell expressing GFP-Atb2p and Sad1p-mEGFP. Dashed outline, cell. Arrowheads, depolymerizing +tip of lPAA microtubules. Arrows, depolymerizing +tip of lPAA microtubules. (**F**,**G**), Timelapse micrographs of cells expressing mEGFP-Myp2p and Ase1p-mCherry. Dashed outline, cell. Red box, position of the contractile ring. F. Appearance of Ase1p soon after Myp2p appearance in the contractile ring. G. More robust Ase1p signal in a mature and constricting contractile ring. (**H**) Confocal fluorescence micrographs of fields of cells expressing GFP-Atb2p. Arrows point to PAA microtubules in a wild-type cell (left) and in *Δase1* cells (right). Red asterisk, dead cell. Dashed outlines, cells. (**I**), Timelapse micrographs of cells expressing GFP-Atb2 in an otherwise wild-type (top) or *Δase1* background (bottom). Dashed outlines, cells. All micrographs are shown as inverted grayscale LUT except for merged images.

**Figure 3 cells-12-00917-f003:**
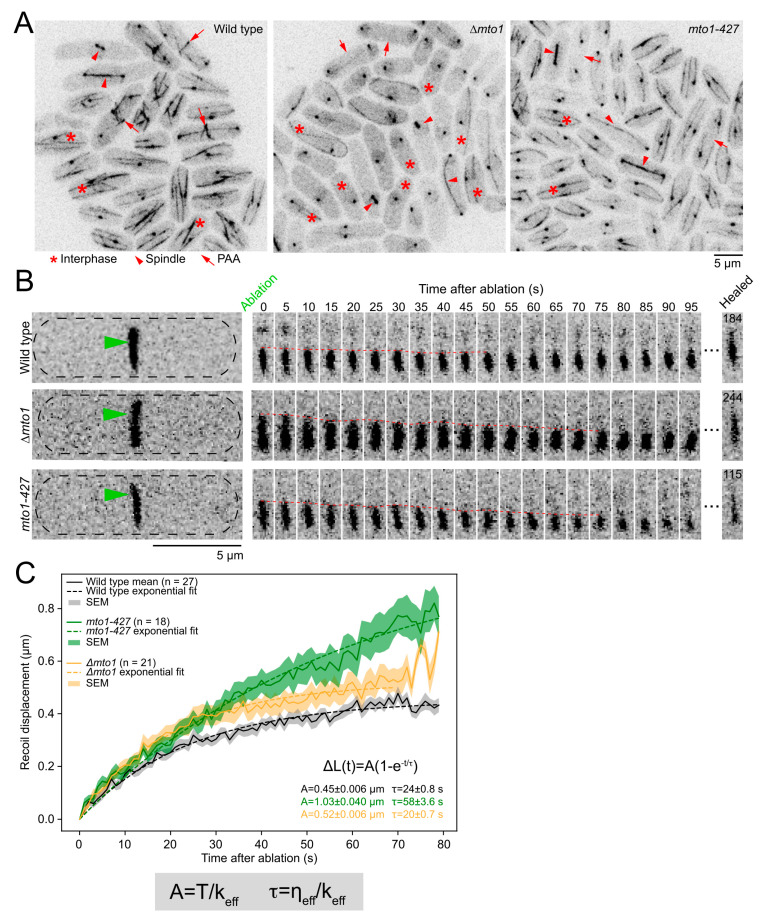
PAA microtubules impart stiffness to the contractile ring. (**A**), Confocal fluorescence micrographs of fields of cells labeled as GFP-Atb2p and Sad1p-mEGFP in wild-type, *∆mto1*, and *mto1-427* cells. (**B**), Time-lapse montages of wild-type (top), *∆mto1* (middle), and *mto1-427* cells (bottom) before and after laser ablation. Green arrowheads, sites of ablation. Dashed red line, position of the severed tip over time. Time of healing after laser ablation is listed for each genotype inside the frame of the micrograph. (**C**), Graph of the recoil displacement of severed tips in wild-type, *∆mto1*, and *mto1-427* cells. Parentheses, number of cells analyzed. All micrographs are shown as inverted grayscale LUT.

**Figure 4 cells-12-00917-f004:**
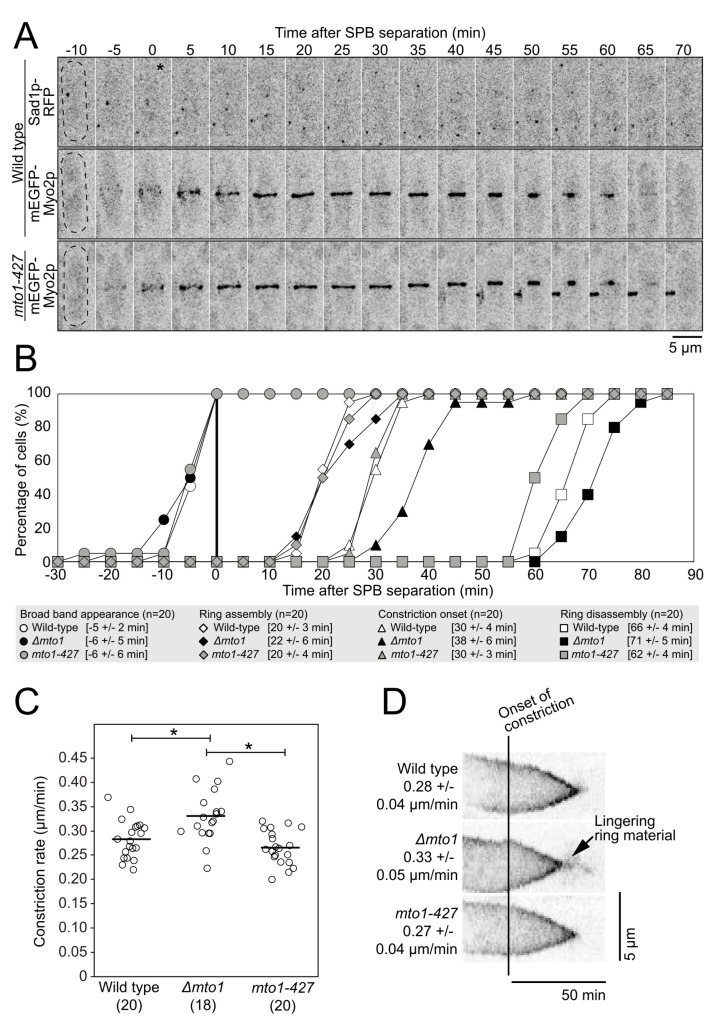
PAA microtubules do not impact the timing of cytokinesis. (**A**), Timelapse micrographs of wild-type and *mto1-427* cells showing the progression of cytokinetic events over time. Asterisk, SPB separation. (**B**), Outcome plot showing the timing of contractile ring events in wild-type, *∆mto1*, and *mto1-427* cells. Square brackets, means ± standard deviations. (**C**), Swarm plot of the constriction rate for individual contractile rings. Bars, means. Asterisk, *p* < 0.05 by Student’s t test. (**D**), Kymographs of wild-type, *∆mto1*, and *mto1-427* contractile rings aligned at the onset of ring constriction. Parentheses, number of cells analyzed. All micrographs are shown as inverted grayscale LUT.

**Figure 5 cells-12-00917-f005:**
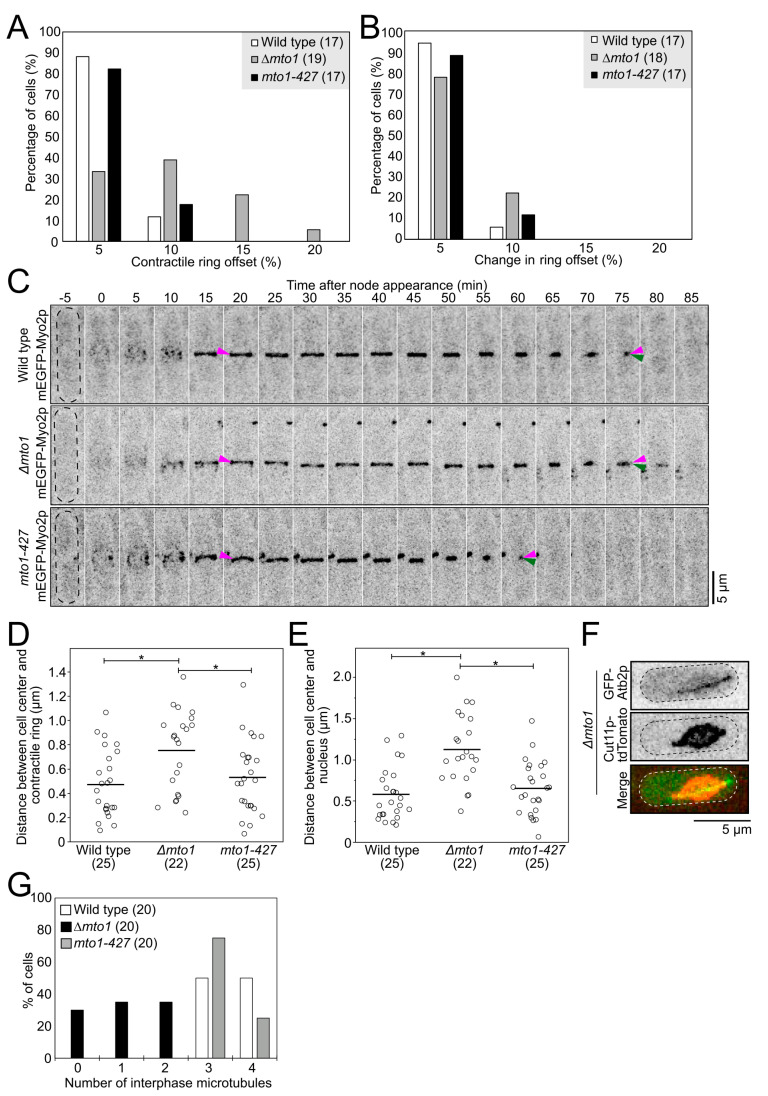
PAA microtubules do not anchor the contractile ring under normal conditions. (**A**), Graph of the ring offset measured at the time of the onset of ring constriction. (**B**), Graph of the change in the ring offset between the time of ring assembly and ring constriction onset. (**C**), Timelapse micrographs of wild-type, *∆mto1*, and *mto1-427* cells showing the position of the ring over time. Dashed outlines, cells. Magenta arrowheads, position of ring at the time of assembly. Green arrowheads, position of ring near the end of constriction. (**D**,**E**), Swarm plots of the distance between the cell center and contractile ring (**D**) and the cell center and nucleus (**E**) in wild-type, *∆mto1*, and *mto1-427* cells. Asterisks, *p* < 0.05 by Student’s *t*-test. (**F**), Representative micrographs of *∆mto1* cells with a lemon-shaped nucleus during interphase. Dashed outlines, cells. (**G**), Graph of the number of interphase microtubules observed in wild-type, *∆mto1*, and *mto1-427* cells. Parentheses, number of cells analyzed. All micrographs are shown as inverted grayscale LUT, except for the merged image.

## Data Availability

Data are contained within the article.

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
