# Peer review of "Actin–Microtubule Crosstalk Imparts Stiffness to the Contractile Ring in Fission Yeast"

_cells, 2023, doi:10.3390/cells12060917_

Round 1

Reviewer 1 Report

The manuscript of Bellingham-Johnstun et al. entitled “Actin-microtubule crosstalk imparts stiffness to the contractile ring in fission yeast” addresses the problem of the interaction between microtubules and microfilaments during cell division using fission yeast as a model system. By Spinning-Disk Confocal Microscopy the AA follow the dynamic in vivo of the post-anaphase array of microtubules and the contractile ring assembly. From quantitative analysis of the cytoskeletal elements and laser ablation of contractile rings, the AA propose that the post-anaphase microtubules do not anchor the ring at the division plane  but may utilize the contractile ring for their assembly. The contractile ring may not receive any benefit from these interactions. In my opinion this manuscript is interesting and deserves publication. I have only a few comments that could be addressed before publication.

l.43 – I think that the AA can consider reviews as Verma et al., 2019 or Sechi et al., 2022 to address the interaction between microtubules and cell cortex during the assembly of the contractile ring.

l.74 – “the polygon is composed of bundles of ~8 microtubules”. These bundles are obvious in cartoon of 2D but not from the images reported through the result section. Please put arrows.

l.162 – “PAA microtubules form a polygon”. This is very interesting. But it is not clear how such a distribution of microtubules can be achieved. How can they be nucleated?

Figure 4. D is lacking. There are two B

Just for my curiosity. The authors localize actin filaments with mCherry-CHD. Why don't they also locate microtubules and microfilaments directly on fixed dividing cells? In this case, it will be impossible to follow the dynamics but the relative positions of the cytoskeletal elements will be more distinct, as for example the respective localization of the contractile ring and microtubules.

Author Response

We thank the review for their thoughtful comments and for their careful reading of the manuscript. We addressed all the comments by entering the necessary changes in the text and figures as requested in this document.

l.43 – I think that the AA can consider reviews as Verma et al., 2019 or Sechi et al., 2022 to address the interaction between microtubules and cell cortex during the assembly of the contractile ring.

We added both references to the text as suggested.

l.74 – “the polygon is composed of bundles of ~8 microtubules”. These bundles are obvious in cartoon of 2D but not from the images reported through the result section. Please put arrows.

We added arrows pointing to the ePAA polygon in Figure 2B. We added the following to the legend of Figure 2B “Arrows point to the ePAA polygon in the first 4 timepoints. The ePAA microtubules cannot be confidently distinguished from the lPAA microtubules in the last 3 timepoints owing to their overlap in projected images.”

In Figure 2C, we relabeled the arrow pointing to the ePAA microtubules to say “ePAA polygon”.

l.162 – “PAA microtubules form a polygon”. This is very interesting. But it is not clear how such a distribution of microtubules can be achieved. How can they be nucleated?

We clarified our interpretation in the results with the following sentence” “Presumably, the eMTOCs that nucleate the microtubules that form the sides of the polygon localize at the vertices that connect the sides of the polygon.”

In addition, the following sentences that summarize our interpretations are in the Discussion: “Presumably, microtubules polymerize from Mto1p/eMTOCs and are rapidly crosslinked by Ase1p into a polygon of ePAA microtubules. Microtubule bundles thus form the sides of the polygon and connect at vertices where the Mto1p/eMTOCs that polymerized those microtubules are likely enriched.”

Figure 4. D is lacking. There are two B

We fixed the figure legend.

Just for my curiosity. The authors localize actin filaments with mCherry-CHD. Why don't they also locate microtubules and microfilaments directly on fixed dividing cells? In this case, it will be impossible to follow the dynamics but the relative positions of the cytoskeletal elements will be more distinct, as for example the respective localization of the contractile ring and microtubules.

This is a great question. Fixation in fission yeast unfortunately does not preserve well the native organization of the cytoskeleton. The actin in the ring often appears wavy, which is never observed in live cells either by confocal or super resolution microscopy. It is possible that this outcome is due to fixation artefact. For this reason, we favour live cell imaging.

Reviewer 2 Report

This paper presents some beautiful and important observations on the post-anaphase array (PPA) of microtubules (MTs) that forms at the site of cell cleavage in the fission yeast, S. pombe. The authors use excellent fluorescence microscopy to observe living cells as they complete mitosis, form their cleavage ring (CR), generate the PPA, and complete the cell division process.  Their ability to view these cells along their long axis, so the CR lies in the plane of focus, allows them to visualize the PPA more clearly than ever before, and their use of multiple fluorescent proteins, assembled as chimeras with several relevant proteins, reveals positional and temporal relationships among these proteins as the cell completes its growth and division cycle.  All this lovely imaging makes the paper a good read and a real contribution to the literature. 

The goal of the work described is to investigate functional relationships between the PPA and the CR. Here the paper is weaker, partly because the effects of these structures on one another appears to be weaker than one might have expected.  The CR will form and function without the PPA, although its behavioral details seem slightly altered. Clearly, however, proteins that associated with myosin in the CR are instrumental in initiating the MTs of the PPA. Moreover, the paper uses its excellent imaging to demonstrate the relative times at which multiple components enter the structures of the CR and PAA, thereby showing that some chains of causality are possible whereas other that are not.

An experiment designed to probe the interaction between myosin and the MT initiating “Mto” proteins suggests to the authors that these proteins do not interact directly, but I believe an alternative interpretation is possible. If the amount of Mto is small relative to the amount of myosin (consistent with the spotty localization of Mto on the CR), and if the binding between myosin and Mto is strong, then doubling the amount of myosin would not be expected to produce a substantial increase in the amount of Mto at the CR. Could the authors use images already in hand to evaluate the amount of Mto that is free in solution when some is CR bound and test whether this suggestion is valid?

The experiments described use either a deletion of Mto1 or an allele identified in the Sawin lab to eliminate the MTs of the PPA and then look for changes in the CR and its properties.  The assumption is that in cells carrying these mutant forms of Mto, the PPA is completely absent.  This is never shown in the current paper, and it is a key point, so I would encourage the authors to add a supplementary figure that demonstrates the legitimacy of this assumption.

The authors make a nice distinction between the stable, Ase1-stabilized MTs of the PPA at the CR plane and the MTs that project from that plane towards both ends of the cell.  They mentioned the dynamics of these MTs and say they suggest that MT plus ends are distal to the PPA, but no data are presented.  I presume they have information on growth and shortening rates plus catastrophe frequencies, and these should be added to support the assertion of MT polarity.

This reader found numerous small issues that need attention to help make the paper easier to read and in some cases more accurate; these are entered as comments on the PDF of the paper that I return to the editorial office.  (I think this is the most efficient way of getting these many comments into the hands of the authors.)  Several more general comments can, however, be expressed here.

I am not convinced by the data in this paper or the authors’ two previously published papers on closely related subjects that the laser ablation of the CR and its subsequent shape changes and repair are doing what the authors think. I have been unable to find published results to convince me that the laser pulses used to “cut” the CR are really doing that. No one has shown, for example, that after ablation as seen by the disappearance of fluorescent myosin the microfilaments of the ring are actually gone in the irradiated area. The “ablation” could be a photobleaching event, rather than a cut.  The first work on this approach that I know of (Silva et al., JCB, 2016) shows convincing mechanical evidence for a cut, because their en-face view of the CR as it is ablated show the ends moving outward (not circumferentially as seen here). The motions described in that paper could be responses to the energy of the laser pulses with subsequent local heating; they certainly don’t look like recoils.  Also, they are faster than the motions seen here.  Thus, I think the “cuts” and subsequent end motions described here could be results of fiber depolyermization, rather than elastic recoil.  It might be worth while for the authors to mention this alternative interpretation of their observations, unless they have reasons to dismiss the idea that are not yet stated in the paper.  In my view, however, the paper stands as a good set of observations, regardless of the interpretation of their ablation experiments.
